# The Prevalence and Predictors of Hypertension and the Metabolic Syndrome in Police Personnel

**DOI:** 10.3390/ijerph18136728

**Published:** 2021-06-22

**Authors:** James D. Yates, Jeffrey W. F. Aldous, Daniel P. Bailey, Angel M. Chater, Andrew C. S. Mitchell, Joanna C. Richards

**Affiliations:** 1Institute for Sport and Physical Activity Research, University of Bedfordshire, Bedford MK41 9EA, UK; jeffrey.aldous@beds.ac.uk (J.W.F.A.); daniel.bailey@brunel.ac.uk (D.P.B.); angel.chater@beds.ac.uk (A.M.C.); andrew.mitchell@beds.ac.uk (A.C.S.M.); 2Sedentary Behaviour, Health and Disease Research Group, Brunel University London, Uxbridge UB8 3PH, UK; 3Division of Sport, Health and Exercise Sciences, Department of Life Sciences, Brunel University London, Uxbridge UB8 3PH, UK

**Keywords:** hypertension, metabolic syndrome, stress, sitting time, police

## Abstract

Hypertension and metabolic syndrome (METSYN) are reportedly high in police forces. This may contribute to health deterioration and absenteeism in police personnel. Police forces comprise of staff in ‘operational’ and ‘non-operational’ job types but it is not known if job type is associated to hypertension and METSYN prevalence. This study aimed to explore the prevalence of hypertension and METSYN, the factors associated with the risk of hypertension and METSYN, and compare physiological, psychological, and behavioural factors between operational and non-operational police personnel. Cross-sectional data was collected from 77 operational and 60 non-operational police workers. Hypertension and METSYN were prevalent in 60.5% and 20% of operational and 60.0% and 13.6% of non-operational police personnel, respectively (*p* > 0.05). Operational job type, moderate organisational stress (compared with low stress) and lower high-density lipoprotein cholesterol were associated with lower odds of hypertension, whereas increasing body mass index was associated with increased odds of hypertension (*p* < 0.05). None of the independent variables were significantly associated with the odds of METSYN. Operational police had several increased cardiometabolic risk markers compared with non-operational police. Given the high prevalence of hypertension and METSYN in operational and non-operational personnel, occupational health interventions are needed for the police and could be informed by the findings of this study.

## 1. Introduction

The number of employees in the English and Welsh police forces is currently 7% lower than in 2010 [1]. With less police officers available for tasks, this may impact on the health and wellbeing of police personnel, increasing job demands and strain, potentially leading to increased sickness absence [2]. Absenteeism is a major problem in the police workforce in England and Wales with 58% of staff reporting one or more days of sickness absence over a 12-month period [3].

A territory police organisation employs personnel in ‘non-operational’ and ‘operational’ job types to effectively serve a community [4]. Operational personnel are required to urgently and dynamically mobilise in response to unpredictable frontline incidents [5]. These personnel have an increased risk of cardiovascular events than the general population due to high levels of cardiometabolic risk markers [5] and sudden unpredictable bursts of strenuous activity [5]. Low physical activity levels, poor diet, and shift work may also contribute to increased cardiovascular risk in these personnel [6]. The presence of hypertension and metabolic syndrome (METSYN), which is a clustering of cardiometabolic risk markers, are highly prevalent in the police [7] and are significant predictors of cardiovascular events in this population [5].

Non-operational personnel work involves being sedentary while completing administrative and desk-based duties in an office environment [4]. High volumes of sedentary behaviour are associated with hypertension [7] and METSYN in the general population [8] and may therefore contribute to increased cardiovascular risk in non-operational personnel [9]. The prevalence of hypertension and METSYN, and differences in physiological, psychological and behavioural factors associated with cardiometabolic disease risk, has not been compared between operational and non-operational police personnel. It is also not known which physiological, psychological, and behavioural factors are associated with the presence of hypertension and METSYN in this occupational group. Research is needed in these areas to help identify if interventions need to be targeted to specific job types in the police and to identify the physiological, psychological, and behavioural factors that should be considered to reduce the risk of hypertension and METSYN. 

The aims of this study were to (a) explore the prevalence of hypertension and METSYN in operational and non-operational police personnel, (b) compare physiological, psychological, and behavioural factors associated with cardiometabolic disease risk between operational and non-operational police personnel, and (c) investigate the association of physiological, psychological, behavioural, occupational, and demographic factors with the presence of hypertension and METSYN in the police.

## 2. Materials and Methods

### 2.1. Study Overview

This study had a cross-sectional design. Participants provided informed consent before data collection. The sample was divided into ‘operational’ (*n* = 77) and ‘non-operational’ (*n* = 60) roles. Operational personnel comprised of ranked police officers and personnel completing ‘policing’ tasks necessary in crime prevention and law enforcement such as scenes of crimes officers, investigation officers, and detention officers [10]. These roles involve large amounts of desk work but also involve mobilising to perform duties and activities that occur outside of the office environment [11]. Non-operational personnel support the organisation in largely administrative capacities [4] and are inclusive of control room, finance, human resources and communication officers, and are exclusively based in office environments.

### 2.2. Participants and Recruitment

A sample of 137 police officers and police staff, representing approximately 6% of the territory police organisation, were recruited via self-selection from two sites in a single territory police organisation in South East England. Recruitment was via site visits, presentations to the workforce, invitation emails from the Police and Crime Commissioner’s office, and adverts on the organisation’s intranet. Participants could be any individual employed by the police organisation free from significant injury or other contraindications to performing the tests for the study.

### 2.3. Testing Protocol and Classifications

#### 2.3.1. Online Survey

Participants initially completed online questionnaires in private to collect demographic information (occupation, length of service, age, and gender); medical history and medication use (blood pressure and cholesterol medication) using non-validated items; behavioural questionnaires including a validated domain specific sitting time questionnaire [12] to measure total daily sitting time and workplace sitting time per day (intraclass correlation coefficients of 0.62–0.86); the validated short form International Physical Activity Questionnaire (IPAQ) (Spearman’s *p* clustered around 0.3) to classify those as low, moderate, or highly active [13]; self-reported smoking status (non-smoker, daily smoker or smoking less than daily) using non-validated items; the validated Alcohol Use Disorder Identification Test-Consumption Questions (AUDIT-C) (AUDIT and it’s versions exhibit Cronbach’s Alpha coefficients between 0.83 and 0.94) [14]; self-reported average hours of sleep per night using a non-validated item. Participants also completed questionnaires assessing psychological variables: the validated Police Organisational Stress (PSQ-ORG) and Police Operational Stress (PSQ-OP) questionnaires (Cronbach’s Alpha > 0.8) [15] to classify those with low, moderate, or high stress [16]. These questionnaires were completed using Qualtrics (Qualtrics, Inc., London, UK).

#### 2.3.2. Physical Data Collection Session

Following completion of online questionnaires, participants were required to either watch a familiarisation video of tests prior to their data collection session or complete the familiarisation in person at their worksite approximately two months prior to data collection. Participants were instructed to attend the data collection session having fasted for ≥9 h, hydrated, and having avoided caffeine intake and strenuous exercise on the day of the data collection session. Tests were completed in the order described below.

##### Blood Biomarkers

Finger prick capillary blood sampling was used to collect 40 μL of whole blood for analysis of total cholesterol (TC), high-density lipoprotein cholesterol (HDL-C), triglycerides, TC:HDL ratio, and blood glucose. Blood was analysed using the Cholestech LDX Analyzer (Cholestech Corp., Hayward, NJ, USA).

##### Blood Pressure

An Omron, M5-I automatic blood pressure monitor (Omron Matsusaka Co Ltd., Matsusaka, Japan) was used to measure resting systolic (SBP) and diastolic (DBP) blood pressure. Participants were seated and rested for 5 minutes prior to the measurement being taken on the left arm. Two measurements were taken with 30 seconds rest between readings. The average of the two readings was recorded.

##### Anthropometry and Body Composition

For anthropometry and body composition measurements, participants removed footwear and personal artefacts. Body mass and body fat % were measured using a bioelectrical impedance analysis device (Tanita BC41MA Segmental Body Composition Analyzer; Tanita Corp., Tokyo, Japan) whilst wearing minimal clothing (shorts and t-shirt). Body mass index (BMI) was calculated as mass (kg)/height (m^2^). Waist circumference (WC) was measured at the mid-point between the lower border of the ribs and the top of the iliac crest [17] using an adjustable tape measure (HaB Direct, Southam, UK). Hip circumference was measured at the widest part of the buttocks [17]. Waist-to-hip ratio was calculated as waist (cm)/hip (cm) circumference.

##### Lung Function

An electronic MicroPlus spirometer (Carefusion, San Diego, CA, USA) was used to measure Forced Vital Capacity (FVC), Forced Expiratory Volume in one second (FEV_1_) and FEV_1_:FVC ratio. Each participant was asked to inspire fully, then exhale as hard and as fast as possible into the microspirometer and keep going for as long as possible. A minimum of three attempts were performed with an acceptability criteria being when there was a ≤0.150 L differences between the largest and next largest FVC and FEV_1_ measurements [18].

##### Handgrip Strength

Handgrip strength was measured following the American Society of Hand Therapists protocol [19]. Participants were seated with shoulders adducted and neutrally rotated. The participant was asked to position the forearm in neutral position with the wrist positioned between 0° and 30° dorsiflexion. The participant gripped the Takei 5401 dynamometer (Takei Scientific Instruments Co., Ltd, Yashorida, Japan) maintaining a maximal isometric contraction of the fingers for three seconds [20]. Three tests were performed on the right (GRIPR) and left (GRIPL) hands with one minute rest between attempts. The highest score for each hand was recorded.

##### Aerobic Fitness

The Chester Step Test was conducted according to published protocols [21] using a pre-recorded audio file and 30 cm height box whilst wearing a Polar FS1 heart rate monitor (Polar Electro, Warwick, UK). The participant listened to the audio file instructions and then commenced stepping up and down from the box in time with a metronome beat at a rate of 60 beats per minute (level 1). Each level was two minutes in duration with heart rate and Borg Rating of Perceived Exertion [22] being recorded 5 seconds before the end of each stage. The metronome beat increased by a rate of 20 beats per minute at each stage and the test continued until the participant reached 80% of their maximum predicted heart rate (220-age) or an RPE of ≥14. Maximum oxygen uptake (VO_2max_) was predicted by plotting a line of best fit between the heart rate data points, projecting the line up to maximum heart rate, and estimating the VO_2max_ (mL/kg/min) from the *x*-axis [21].

##### Hypertension and Metabolic Syndrome Definitions

Hypertension was classified as SBP ≥ 130 mmHg and/or DBP ≥ 80 mmHg [23]. METSYN was classified using the International Diabetes Federation definition [24]. This requires the presence of central obesity (WC ≥ 94 cm for males and ≥80 cm for females), plus two or more of the following risk factors: raised blood pressure (SBP ≥ 130 mmHg, DBP ≥ 85 mmHg or using blood pressure medication), reduced HDL-C (<1.03 mmol/L for men and <1.29 mmol/L for women), high triglycerides (≥1.7 mmol/L or using cholesterol medication), and impaired fasting glucose (≥5.6 mmol/L).

### 2.4. Statistical Analysis

Statistical analysis was conducted using SPSS version 22 (IBM, Armonk, New York, NY, USA). Data is presented as mean ± SD unless otherwise stated. Where there was missing data, those cases were excluded listwise from the relevant models. Chi-square tests were used to explore the association of job type with the prevalence of hypertension, METSYN, abdominal obesity, low HDL-C, high triglycerides, impaired fasting glucose, and smoking status. The association of job type with stress and physical activity levels was explored using chi-square tests. Differences in cardiometabolic risk markers, anthropometric variables, lung function, handgrip strength, and aerobic fitness were compared between job types using independent samples *t*-tests.

Binary logistic regression using the enter method was employed to evaluate odds ratios (ORs) and 95% confidence intervals (CI) for having METSYN and hypertension. The following variables were included in the METSYN model: job type, age, sex, length of service, smoking status, operational stress score, organisational stress score, physical activity level, FEV_1_, FVC, VO_2max_, GRIPR, daily sitting and workplace sitting, SLEEP and AUDIT-C score. The hypertension model additionally included TC, HDL-C, triglycerides, glucose, and BMI. Statistical significance was set at *p* ≤ 0.05. Statistical trends in the data were considered apparent when *p* > 0.05–*p* < 0.10.

## 3. Results

Descriptive characteristics of the sample are shown in Table 1. There was no significant difference in age between job types but length of service was significantly longer in operational personnel (*p* = 0.02). Across the whole sample, there was a prevalence of 17.2% for METSYN, 60.3% for hypertension, 50.0% for high WC, 27.9% for low HDL-C, 16.5% for high triglycerides, and 14.0% for impaired fasting glucose. The prevalence of hypertension, METSYN, daily smokers, and the individual METSYN risk factors for each job type is shown in Table 2. Job type was not significantly associated with any of these variables (Table 2). 

Job type was not significantly associated with operational or organisational stress (Table 3). A significant association between job type and physical activity levels was found with a higher proportion of operational personnel engaging in moderate and high levels of physical activity than non-operational personnel (Table 3).

Operational personnel had significantly higher WC, waist-to-hip ratio, FEV_1_, FVC, VO_2max_, and handgrip strength than non-operational personnel. Operational personnel also had significantly lower HDL-C concentrations and engaged in significantly less daily sitting (Table 4). There was a trend for a higher BMI and TC:HDL ratio in operational personnel. There was a trend for higher body fat % and FEV_1_:FVC ratio in non-operational personnel.

The regression model for hypertension explained 58% of variance and correctly classified 83% of cases. Variables contributing significantly to the model were job type, organisational stress, HDL-C, and BMI (Table 5). The odds of hypertension were 0.10 times lower for operational than non-operational personnel, 0.12 times lower for those with moderate organisational stress than those with low organisational stress, 0.16 times lower for each unit increase in HDL, and 1.24 times higher for each unit increase in BMI. There was also a trend for increasing handgrip strength and sleep being associated with higher odds of hypertension.

The regression model for METSYN explained 28% of variance and correctly classified 86% of cases. The odds of having METSYN were not significantly associated with any independent variables. There was a trend for lower odds of METSYN with increasing alcohol use disorder scores.

## 4. Discussion

### 4.1. Overview of Main Findings

The main findings of this study were that the high prevalence of hypertension in police personnel was associated with job type, whereas the prevalence of METSYN was not. Specifically, operational personnel had lower odds of hypertension than non-operational. In addition, operational personnel had higher levels of abdominal adiposity and lower HDL-C, but had more favourable lung function and handgrip strength profiles, and engaged in more physical activity and less daily sitting.

### 4.2. Hypertension

Both operational and non-operational police had a high prevalence of hypertension (60.5% and 60.0% prevalence, respectively) that exceeds the 28% prevalence seen in the general population in England [25]. Our findings also suggest that hypertension may be more prevalent than findings from the Airwave Health Monitoring Study in the UK where prevalence rates of hypertension were 29.5% in police personnel [26]. A study using the same Airwave Health Monitoring Study cohort of police employees by Gibson et al. [27]. also reported high levels of elevated blood pressure in males (68.7%) and females (33.5%), which are more aligned with the prevalence in the current study. The increased prevalence found by Gibson et al. [27] can be explained by the use of thresholds to define elevated blood pressure (SBP ≥ 130 mmHg and/or DBP ≥ 85 mmHg) compared to Elliott et al. [26], who aimed to determine prevalence of hypertension (SBP ≥ 140 mmHg and/or DBP ≥ 90 mmHg). The present study extends knowledge in this field by demonstrating that hypertension is highly prevalent in both operational and non-operational police personnel, which had not previously been reported. 

Although the prevalence of hypertension was high in both operational and non-operational personnel, operational personnel had lower odds of hypertension. This was despite operational personnel having lower HDL-C and a higher abdominal adiposity, which are risk factors for hypertension [28,29]. Operational personnel did possess more favourable grip strength, lung function, cardiorespiratory fitness levels, engaged in higher amounts of physical activity, and also sat less throughout the day than non-operational personnel. These may be protective factors that could explain the lower odds of hypertension in operational personnel [30,31,32,33,34]. However, it should be noted that the prevalence rates of hypertension for non-operational and operational personnel were not different according to the chi-square analysis. This suggests that other factors that were considered statistically significant in the logistic regression model are important when evaluating the likelihood of having hypertension, which is indeed supported by the findings.

Police in operational and non-operational roles engage in large amounts of desk work [4]. This was supported by findings in the present study in which operational and non-operational personnel engaged in 5.6 and 6.3 h per day of workplace sitting, respectively. However, daily and workplace sitting time were not associated with the presence of hypertension. On the contrary, previous prospective evidence in the general population (*n* = 11,837) found that higher total daily sedentary behaviour was associated with an increased risk of hypertension [34]. The smaller sample size and cross-sectional design of the present study could explain the disparity in findings compared to previous research. Prospective studies in larger samples of police personnel are thus needed to confirm whether sedentary behaviour and other demographic, health, and behavioural factors are associated with the high prevalence of hypertension in the police. 

Moderate levels of organisational stress were associated with lower odds of hypertension compared with low organisational stress levels. This is in contrast with a previous systematic review that reported stress was frequently associated with hypertension in police officers [35]. It is not clear why moderate levels of stress could be associated with lower odds of hypertension. Operational and non-operational personnel had similar stress scores so job type may not be an important factor in explaining this phenomenon. Further research is thus required to aid in understanding the health effects of stress in the police.

### 4.3. Metabolic Syndrome

There was a high prevalence of METSYN in non-operational and operational personnel in the present study (13.6% and 20.0%, respectively). This is similar to the general European population with prevalence rates of 10–30% reported [36]. The present findings are also similar to the 16.8% prevalence reported in police officers in India [37]. Previous literature has indicated that greater METSYN prevalence (36.4%) exists in police officers compared with the present study [38]. In the study by Leischik et al. [38], it is unclear whether, like in the current study, a high WC was a mandatory risk factor in their definition of METSYN. If it was not, then this would at least partly explain the higher prevalence of METSYN in their study. Regardless, METSYN prevalence in this occupational group is high and intervention strategies to improve cardiometabolic health are thus warranted.

None of the variables evaluated in this study were significantly associated with METSYN. There was a trend for lower odds of METSYN with increasing alcohol use disorder scores. This is in contrast to previous research that reported higher alcohol intake to be associated with an increased risk of METSYN in the general population [39] and in police officers in China [40]. Although the questionnaires used to measure alcohol intake varied between studies, it is unlikely that this alone would explain these disparate findings. Instead, it may be possible that police personnel with higher alcohol use disorder scores may engage in other healthy behaviours that protect them against abnormal cardiometabolic health. 

It has been suggested that irregular working hours, shift patterns, and diet may be contributing factors to an increased likelihood of METSYN in the police [41,42]. These were not assessed in the present study and should be considered in future research to help elucidate possible links with alcohol intake in relation to the risk of METSYN in this occupational group. Furthermore, previous studies in the police have reported associations between a number of the variables evaluated in this study and METSYN. For instance, age, BMI, and smoking were significantly associated with METSYN in Indian police officers [43]. These contrasting findings may be due to differences in societal health behaviours between India and the UK. It is recommended that research is conducted to identify the predictors of METSYN in the police and how these may differ across different job types and in different environmental contexts.

### 4.4. Physiological, Psychological, and Behavioural Profiles of Operational and Non-Operational Police

Our findings revealed an adverse cardiometabolic health profile in operational compared to non-operational personnel demonstrated by lower HDL-C, higher WC and WHR, and a trend for higher BMI and TC:HDL ratio. To the authors’ knowledge, no previous research has compared cardiometabolic risk between different job types in the police. Previous evidence has consistently documented that police officers have an increased cardiometabolic risk than the general population [7]. This increased physiological risk existed despite operational personnel having significantly better handgrip strength, lung function, higher physical activity levels, and lower daily sitting, which have been favourably associated with cardiometabolic health [30,31,32,33,34]. This supports the need for occupational health interventions to reduce the risk of cardiometabolic disease in the police, especially those in operational jobs.

This study’s novel findings also demonstrate that workplace sitting time is similar between job types in the police, which shows that operational personnel conduct a significant amount of their work while seated. This study did not explore how or where the participants engaged in sitting, such as whether it was predominantly sitting at a desk, while patrolling in police vehicles or conducting tasks like interviews and taking statements. This should be addressed in future research to help inform intervention development and occupational health policies.

### 4.5. Strengths and Limitations

The main strength of this study includes the comprehensive evaluation of a multitude of physiological, psychological, behavioural, and demographic variables in the context of their differences between operational and non-operational police personnel and their association with hypertension and METSYN. The study is limited by not including some factors that could be important in explaining differences in health profiles between operational and non-operational personnel and could have a role in the risk of hypertension and METSYN. This may include shift work, family history of hypertension and METSYN, and diet. These factors should be considered in future research. The study is also limited due to it being set within a specific occupational group and the findings may thus not generalise to other occupation groups or the general population.

## 5. Conclusions

This study suggests that there is a high prevalence of hypertension and METSYN in both operational and non-operational police personnel. It is therefore recommended that interventions to reduce the risk of hypertension and METSYN in the police target both of these job types to reduce the risk of cardiometabolic disease. It should be acknowledged that several cardiometabolic risk markers were higher in operational police, yet an operational job type was associated with lower odds of hypertension. Concerted efforts may thus be needed for both occupational groups.

## Figures and Tables

**Table 1 ijerph-18-06728-t001:** Descriptive characteristics of the sample (*n* = 137).

Variable	Non-Operational (*n* = 60)	Operational (*n* = 77)	Whole Sample
Female (*n*)	49	40	89
Male (*n*)	11	37	48
Caucasian (*n*)	51	70	88.3
Non-Caucasian (*n*)	9	7	11.7
Age (years)	43 ± 12	43 ± 9	43 ± 11
Length of service (years)	11 ± 10	15 ± 8	13 ± 9

**Table 2 ijerph-18-06728-t002:** Associations of job type with the prevalence of hypertension, metabolic syndrome, metabolic syndrome risk factors, and daily smoking (*n* = 137).

Variable	Non-Operational	Operational	*p* Value
Hypertension (%)	60.0	60.5	0.95
Metabolic syndrome (%)	13.6	20.0	0.34
High waist circumference (%)	46.7	52.6	0.49
Low HDL-C	23.7	31.2	0.34
High triglycerides	13.6	18.2	0.47
Impaired fasting glucose	16.9	11.7	0.38
Daily Smoker (%)	8.3	5.3	0.48

Chi-square test used to obtain *p* values. *n* = 134 for Metabolic syndrome. HDL, high-density lipoprotein cholesterol.

**Table 3 ijerph-18-06728-t003:** Associations of job type with stress and physical activity levels.

Variable	Non-Operational			Operational			*p* Value
	Low	Moderate	High	Low	Moderate	High	
Police operational stress score (%) ^a^	51.7	27.6	20.7	55.3	30.3	14.5	0.64
Police organisational stress score (%) ^b^	48.3	33.3	18.3	45.9	28.4	25.7	0.58
Physical activity level classification (%) ^c^	33.3	36.7	30.0	15.8	38.2	46.1	0.04

Chi-square test used to obtain *p* values. *n* = 134 for police operational and organisational stress score, *n* = 136 for physical activity level classification.^. a^ Police operational stress classification: Low <2.6, Moderate 2.7–3.9, High ≥4.0 [16]. ^b^ Police organisational stress classification: Low <2.0, Moderate 2.1–3.4, High ≥3.5 [16]. ^c^ International Physical Activity Questionnaire classification: Low = not meeting any of criteria for Moderate or High, Moderate = 3 or more days of vigorous intensity activity and/or walking of at least 30 min per day or 5 or more days of moderate intensity activity and/or walking of at least 30 min per day or 5 or more days of any combination of walking, moderate intensity or vigorous intensity activities achieving a minimum total physical activity of at least 600 MET-minutes/week, High = Vigorous intensity activity on at least 3 days achieving a minimum total physical activity of at least 1500 MET-minutes/week or 7 or more days of any combination of walking, moderate intensity or vigorous intensity activities achieving a minimum total physical activity of at least 3000 MET-minutes/week [14].

**Table 4 ijerph-18-06728-t004:** Differences in measures of health, health behaviours, and fitness between operational and non-operational police workers (*n* = 137). Data presented as mean ± SD.

Variable	Non-Operational	Operational	*p* Value
Systolic blood pressure (mmHg)	122 ± 14	125 ± 13	0.21
Diastolic blood pressure (mmHg)	81 ± 9	81 ± 8	0.86
Body fat percentage	33.5 ± 8.5	30.8 ± 9.0	0.08
Waist circumference (cm)	82.5 ± 13.5	89.7 ± 13.3	<0.01
Waist-to-hip ratio	0.80 ± 0.07	0.88 ± 0.20	<0.01
FEV_1_ (L)	3.2 ± 0.6	3.6 ± 0.8	<0.01
Forced vital capacity (L)	3.9 ± 0.7	4.5 ± 1.1	<0.01
FEV_1_:FVC ratio (%)	81.6 ± 5.5	79.7 ± 5.7	0.06
Maximum oxygen uptake (mL/kg/min)	33.2 ± 6.4	36.1 ± 6.7	0.01
Grip strength right hand (Kg)	27.2 ± 7.7	36.5 ± 9.3	<0.01
Grip strength left hand (Kg)	25.7 ± 8.2	33.8 ± 8.9	<0.01
Daily sitting (hours/day)	8.4 ± 2.8	6.8 ± 2.8	<0.01
Workplace sitting (hours/day)	6.3 ± 3.0	5.6 ± 2.6	0.11
Sleep (hours/day)	6.5 ± 0.8	6.7 ± 1.1	0.18
Alcohol use disorder score	3.8 ± 2.1	3.5 ± 2.2	0.45
Total cholesterol (mmol/L)	5.38 ± 1.09	5.42 ± 1.16	0.82
HDL-C (mmol/L)	1.57 ± 0.48	1.41 ± 0.43	0.05
TC:HDL ratio	3.73 ± 1.52	4.24 ± 1.83	0.08
Triglycerides (mmol/L)	1.14 ± 0.63	1.28 ± 0.78	0.27
Glucose (mmol/L)	5.01 ± 0.70	5.09 ± 0.46	0.46
Body mass index (kg/m^2^)	26.4 ± 5.4	28.0 ± 4.4	0.06

Independent samples *t*-tests were used to obtain *p* values. FEV_1_, Forced expiratory volume in one second; FEV_1_:FVC ratio, Forced expiratory volume in one second:Forced vital capacity ratio; HDL-C, High density lipoprotein cholesterol; TC:HDL ratio, Total cholesterol:High-density lipoprotein cholesterol ratio. Variables with missing data; Maximum oxygen uptake (*n* = 7), HDL-C (*n* = 7), Triglycerides (*n* = 4) and Body fat % (*n* = 1).

**Table 5 ijerph-18-06728-t005:** Odds ratios for metabolic syndrome and hypertension according to each predictor variable (*n* = 121).

Variable	Metabolic Syndrome		Hypertension	
	OR (95% CI)	*p* Value	OR (95% CI)	*p* Value
Job type ^a^	2.38 (0.58, 9.78)	0.23	0.10 (0.02, 0.51)	0.01
Age (Years)	1.03 (0.94, 1.14)	0.50	1.03 (0.94, 1.13)	0.54
Sex ^b^	2.18 (0.29, 16.42)	0.45	2.71 (0.25, 29.28)	0.41
Length of service (Years)	0.99 (0.91, 1.08)	0.87	1.03 (0.95, 1.12)	0.44
Daily smoker ^c^	1.37 (0.08, 23.17)	0.83	2.94 (0.26, 32.75)	0.38
Police operational stress score (Moderate) ^d^	1.20 (0.32, 4.55)	0.79	1.42 (0.33, 6.04)	0.64
Police operational stress score (High) ^d^	2.17 (0.36, 13.07)	0.40	0.41 (0.07, 2.45)	0.33
Police organisational stress score (Moderate) ^d^	1.06 (0.27, 4.11)	0.93	0.12 (0.03, 0.50)	<0.01
Police organisational stress score (High) ^d^	1.24 (0.25, 6.145)	0.79	0.54 (0.12, 2.50)	0.43
Physical activity level classification (Moderate) ^e^	0.43 (0.10, 1.96)	0.28	0.58 (0.12, 2.73)	0.49
Physical activity level classification (High) ^e^	0.47 (0.10, 2.27)	0.35	0.58 (0.10, 3.46)	0.55
FEV_1_ (L)	8.25 (0.32, 216.30)	0.21	0.35 (0.02, 5.66)	0.46
Forced vital capacity (L)	0.14 (0.01, 1.45)	0.10	1.49 (0.19, 11.36)	0.70
Maximum oxygen uptake (mL/kg/min)	0.94 (0.82, 1.08)	0.12	1.01 (0.89, 1.15)	0.85
Grip strength right hand (Kg)	1.07 (0.98, 1.17)	0.37	1.11 (0.98, 1.26)	0.09
Daily sitting (hours/day)	1.16 (0.86, 1.57)	0.32	1.04 (0.78, 1.39)	0.80
Workplace sitting (hours/day)	0.86 (0.66, 1.12)	0.27	0.91 (0.69, 1.19)	0.49
Sleep (hours/day)	0.83 (0.46, 1.50)	0.54	1.82 (0.95, 3.49)	0.07
Alcohol use disorder score	0.76 (0.57, 1.02)	0.07	1.05 (0.78, 1.40)	0.76
Total cholesterol (mmol/l)	-	-	1.59 (0.82, 3.08)	0.17
High density Lipoprotein-Cholesterol (mmol/L)	-	-	0.13 (0.03, 0.70)	0.02
Triglycerides (mmol/L)	-	-	3.29 (0.52, 20.63)	0.20
Glucose (mmol/L)	-	-	0.66 (0.20, 2.19)	0.50
Body mass index (kg/m^2^)	-	-	1.24 (1.03, 1.48)	0.02

Binary logistic regression using enter method was used to obtain *p* values. FEV_1_, Forced expiratory volume in one second. ^a^ Non-operational personal used as reference category, ^b^ Females used as reference category, ^c^ Non-smoker used as reference category, ^d^ Low stress scored used as reference category, ^e^ Low physical activity levels used as reference category.

## Data Availability

The unidentified data presented in this study is available on reasonable request from the corresponding author.

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
