# Peer review of "The Prevalence and Predictors of Hypertension and the Metabolic Syndrome in Police Personnel"

_ijerph, 2021, doi:10.3390/ijerph18136728_

Round 1
Reviewer 1 Report
Yates et al in their article, “The prevalence and predictors of hypertension and the metabolic syndrome in police personnel”, explore the prevalence of hypertension and metabolic syndrome in police personnel, by dividing them into two groups: operational and non-operational. The authors report some interesting findings in their study, such as the prevalence of hypertension is less in the operational group than in the non-operational group. Whereas, surprisingly the prevalence of the metabolic syndrome is more in the operational group than the non-operational group. Though the data wasn’t significantly significant. The study suggests that there is a high prevalence of hypertension and metabolic syndrome in both operational and non-operational police personnel, and this should help the policymakers and health workers to focus their attention when developing occupational health programs for the police.
Comments:
- The authors also do not mention the racial and ethnic profile of the police personnel inducted in the study. It would be interesting to see how the prevalence of hypertension and metabolic syndrome varies between racial/ethnic groups.
- I was wondering why the odds ratio are not computed/provide for Metabolic syndrome and variables such as the total cholesterol (mmol/l), high-density Lipoprotein-Cholesterol (mmol/L), triglycerides (mmol/L), glucose (mmol/L), and body mass index
Author Response
Response: We thank the reviewer for taking the time to review our manuscript and for their positive comments. We have responded to your comments below in red.
1. The authors also do not mention the racial and ethnic profile of the police personnel inducted in the study. It would be interesting to see how the prevalence of hypertension and metabolic syndrome varies between racial/ethnic groups.
Response: The ethnic profile of the sample is now presented in Table 1. The aim of the study was to compare operational and non-operational personnel and thus the comparison of ethnic groups has not been included. Furthermore, there would be insufficient statistical power due to the low number of participants in the non-white ethnic group.
2. I was wondering why the odds ratio are not computed/provide for Metabolic syndrome and variables such as the total cholesterol (mmol/l), high-density Lipoprotein-Cholesterol (mmol/L), triglycerides (mmol/L), glucose (mmol/L), and body mass index.
Response: Odds ratios for the presence of metabolic syndrome have been presented in Table 5. Odds ratios for total cholesterol, high-density Lipoprotein-Cholesterol, triglycerides, glucose, and body mass index predicting metabolic syndrome have not been analysed as these variables are criterion variables for classifying the presence of the metabolic syndrome. It’s considered that it would not be statistically acceptable to include said variables in both the independent and dependent variables.
Reviewer 2 Report
The authors aimed to explore combination of hypertension and metabolic syndrome in police officers. The manuscript is well written and fully intelligible. However there are some major issue that need to be addressed.
1- in both population studied the authors reported only the percentage of metabolic syndrome. However, they indicated in method what data were used to achieve this diagnosis. I guess they will have no problem to report each of the single diagnosis (diabetes, impaired fasting glucose, dyslipidemia, obesity) in both the studied population.
2- In table 2 they indicated that medication used for hypertension was extremely lower compared to diagnosis. If I just compare the two percentage it is obvious that at least a mistake in making table was done. Moreover, it should be interesting to know the number of medication and, if possible, the main class of drugs used, comparing the two groups.
3- The data in table 4 are not clear. Moreover, these data did not appear to be accordance to table 2. In fact, if these data are correct it is not fully intelligible how such an higher rate of hypertension or metabolic syndrome was found.
4- The authors reported that in the study by Elliot there was a lower incidence of hypertension in female police officers. However, just the next sentence they declared these data may not to be compared to this study due they did not performed an analysis in percentage according to gender differences. In addition, in table 5 there was an OR of about 2. I suggest to increase the value of the manuscript performing this analysis for both hypertension and metabolic syndrome, in order to have a better comparison to previous literature data.
5- I did not understand OR in hypertension. If "non-operational" is the reference category, considering the difference in diagnosis percentage reported in table 2, it is no possible that operative officers may present an increased risk.
6- binary variables (such as sex or job type) OR is evaluated using logistic binary regression. However, some other variables (such as blood values of HDL or Maximum oxygen up-take) that are not binary. How the author performed a binary logistic for these parameters? Were them categorized?
7- Due many times they mention a possible "statistical trend", I suggest to include a statement about p values they established to be evaluated as trend in methods
As minor issue in line 274 I did not understand what 4 stands for. Please address.

Author Response
We thank the reviewer for taking the time to review our manuscript and for their positive comments. We have responded to your comments below in red.
1- in both population studied the authors reported only the percentage of metabolic syndrome. However, they indicated in method what data were used to achieve this diagnosis. I guess they will have no problem to report each of the single diagnosis (diabetes, impaired fasting glucose, dyslipidemia, obesity) in both the studied population.
Response: The prevalence of each metabolic syndrome criterion is now provided on lines 193-196. The association of job type with the presence of each metabolic syndrome criterion variable has also been added to Table 2.
2- In table 2 they indicated that medication used for hypertension was extremely lower compared to diagnosis. If I just compare the two percentage it is obvious that at least a mistake in making table was done. Moreover, it should be interesting to know the number of medication and, if possible, the main class of drugs used, comparing the two groups.
Response: The number of individuals medicated for high blood pressure was indeed small. Being on such medication would require the participants to have had a previous medical diagnosis of hypertension for which their clinician deemed appropriate to prescribe medication. It is thus likely that a large proportion of participants with hypertension according to the criteria used in this study had not been diagnosed with high blood pressure or had been prescribed diet or lifestyle changes instead of medication. Indeed, it is estimated that a large proportion (nearly half of all adults) of the population have undiagnosed hypertension (Tapela et al., 2021. Prevalence and determinants of hypertension control among almost 100 000 treated adults in the UK. Open heart, 8(1), p.e001461). This same response also applies to the cholesterol medication. As there were such small numbers of individuals using cholesterol and blood pressure medication we have now removed the chi-square analyses for these variables as there was an n=0 for some cross tabulated groups which thus renders the analysis not useful. The class of drugs was not collected in this study.
3- The data in table 4 are not clear. Moreover, these data did not appear to be accordance to table 2. In fact, if these data are correct it is not fully intelligible how such an higher rate of hypertension or metabolic syndrome was found.
Response: Thank you for noticing this error in our data. We have now thoroughly checked our data and noticed some inconsistencies and have corrected these issues, and conducted all of our analyses again. This has led to some changes in our findings as per the highlighted text in the results. The corresponding parts of the discussion have been amended in light of the change in the findings.
4- The authors reported that in the study by Elliot there was a lower incidence of hypertension in female police officers. However, just the next sentence they declared these data may not to be compared to this study due they did not performed an analysis in percentage according to gender differences. In addition, in table 5 there was an OR of about 2. I suggest to increase the value of the manuscript performing this analysis for both hypertension and metabolic syndrome, in order to have a better comparison to previous literature data.
Response: The aim of the present study was to compare prevalence between operational and non-operational personnel. The study by Elliott et al. (2014) did not conduct such a comparison. To better align the discussion with aims of our study, lines 267-269 have been amended to discuss prevalence rates of hypertension and elevated blood pressure in the whole sample of Elliott et al. (2014). We have then removed the suggestion that it is not possible to compare our findings to Elliott et al.
5- I did not understand OR in hypertension. If "non-operational" is the reference category, considering the difference in diagnosis percentage reported in table 2, it is no possible that operative officers may present an increased risk.
Response: From our updated analysis, the prevalence of hypertension was similar between operational and non-operational (Table 2). The mean systolic and diastolic blood pressure values in Table 2 also show no difference between job types, but a large SD suggesting there is indeed many individuals who may be classified as having high blood pressure or hypertension. However, the odds of having hypertension were lower in operational than non-operational personnel when other health and behavioural variables had been factored into the predictive regression model (Table 5). We have included a discussion around this on lines 280-292.
6- binary variables (such as sex or job type) OR is evaluated using logistic binary regression. However, some other variables (such as blood values of HDL or Maximum oxygen up-take) that are not binary. How the author performed a binary logistic for these parameters? Were them categorized?
Response: In binary logistic regression, although the dependent variable must be categorical, the independent variables may be either categorical or scale. Thus, in this analysis, the ORs for continuous independent variables reflect the OR of the dependent variable for a 1-unit increase in the independent variable.
7- Due many times they mention a possible "statistical trend", I suggest to include a statement about p values they established to be evaluated as trend in methods.
Response: thank you for this suggestion. This has been clarified on lines 187-188.
8- As minor issue in line 274 I did not understand what 4 stands for. Please address.
Response: This was incorrect formatting of a reference and has been corrected on line 293.
Round 2
Reviewer 2 Report
The authors made a deep revision of the data, the results and the manuscript.
Now it sound more scientific than was before.
Some issue are still present and need to be addressed
1- the authors reported that medication used were recorded. They removed the comparative data from table 1 but leave a statement about medication in first paragraph of results. If no comparison is made between the % of medication use I did not understand what is the meaning to indicate it.
2- in line 192-195 the authors reported many percentage but it not indicated what group is referring to. Please address
Finally I suggest the authors to comment a the low medication percentage compared to high hypertension diagnosis they described including a possible non-pharmaceutical intervention as explanation. Indeed, their finding is mostly related to the ACC/AHA low cut-off used.
Author Response
Thank you for reviewing our manuscript and providing feedback. Our responses to comments can be found below, highlighted in red. Our updated manuscript has tracked changes and highlighted text where edits have been made.
1-the authors reported that medication used were recorded. They removed the comparative data from table 1 but leave a statement about medication in first paragraph of results. If no comparison is made between the % of medication use I did not understand what is the meaning to indicate it.
Response: Thank you for the comments. We have removed the % of medication use from first paragraph of results on line 195.
2- in line 192-195 the authors reported many percentage but it not indicated what group is referring to. Please address
Response: This was referring to prevalence across the whole sample. Line 193 has now been amended to state this for clarity.
3- Finally I suggest the authors to comment a the low medication percentage compared to high hypertension diagnosis they described including a possible non-pharmaceutical intervention as explanation. Indeed, their finding is mostly related to the ACC/AHA low cut-off used.
Response: Thank you for the suggestion. In line with the reviewer’s first suggestion, we have now removed reporting of medication use prevalence in the results. As the manuscript now does not report any medication use statistics, it does not seem appropriate to discuss anything in relation to medication use in the discussion.